# Comparison Between In-Office Versus Remote Follow-Up Costs in Patients with Pacemakers and Reimbursed Transportation in a Portuguese District Hospital

**DOI:** 10.3390/healthcare13243257

**Published:** 2025-12-12

**Authors:** João Oliveira, Sandra Oliveira, Vítor Martins, Cristina Reis, Patrícia Branco, Helena Pedrosa, Luís Casalta, Tânia Parreira

**Affiliations:** 1School of Management and Technology, Santarém Polytechnic University, 2001-904 Santarém, Portugal; sandra.oliveira@esg.ipsantarem.pt; 2Santarém District Hospital, 2005-177 Santarém, Portugal; vitor.martins@ulsleziria.min-saude.pt (V.M.); cristina.reis@ulsleziria.min-saude.pt (C.R.); patricia.matias@ulsleziria.min-saude.pt (P.B.); helena.pedrosa@ulsleziria.min-saude.pt (H.P.); luis.casalta@ulsleziria.min-saude.pt (L.C.); tania.parreira@ulsleziria.min-saude.pt (T.P.)

**Keywords:** Portuguese district hospital, remote monitoring, pacemakers, national health service, non-urgent patient transportation

## Abstract

**Background:** Digital technologies can contribute to healthcare democratization in an ethical, safe, and sustainable context. The remote monitoring of implantable cardiac devices enables the follow-up of patients from a distance. Patients with anti-bradycardia pacemakers represent the largest group and have the least access to this technology due to the controversial cost–benefit ratio and barriers to its widespread implementation, such as equipment costs and organizational challenges. In contrast, reimbursed transportation in Portugal reached approximately 82 million euros in 2024. **Objectives**: The aim of the present study was to assess the financial viability of remote pacemaker follow-up in a Portuguese district hospital, comparing the non-urgent transportation costs and the investment in remote monitoring equipment while measuring user acceptance and satisfaction. **Methods**: A total of 41 surveys were conducted with patients who received a pacemaker and used publicly reimbursed transportation. The projected costs were calculated for two simulated protocols: the first involved in-person visits every six months, while the second involved in-person visits every two years with remote consultations every six months, over the expected lifespan of the devices. EZR, version 1.61, was used. **Results**: Our data showed a 74% overall reduction in face-to-face visits. The implementation of remote follow-up would result in a cost saving of EUR 373/patient (21.2%), with total reimbursement (*p* = 0.01151). The savings increased to 33.3%, reaching EUR 764/patient (*p* = 0.0002742) for distances greater than 60 km (round trip) for ambulance users with total reimbursement. Acceptance and satisfaction achieved 88%. **Conclusions**: Remote monitoring of pacemakers can be a financially viable alternative with high acceptance and satisfaction.

## 1. Introduction

Remote monitoring of implantable cardiac devices is a technology that has been available since 2001. It allows pacemakers, event recorders, implantable cardioverter defibrillators (ICDs), and cardiac resynchronization devices (CRTs) to be remotely monitored, using digital web-based platforms. These platforms can display all data as conventional programmers and maintain the communications history and recordings [1]. The benefits are well established in the literature, particularly in detecting arrhythmic events, managing device malfunctions, and optimizing battery life, reducing hospital visits, admissions, and costs for patients, and improving their quality of life [1,2,3,4,5,6,7,8,9,10,11,12,13,14,15,16,17]. There are still some topics with controversial results, particularly in relation to mortality [10,18,19,20,21,22].

Of all implantable cardiac devices available, anti-bradycardia pacemakers are the most common [23,24,25]. Despite recommendations, users with anti-bradycardia pacemakers have poor access to this technology. In this group, the cost–benefit ratio is less clear [26,27,28]. According to a survey carried out by the European Heart Rhythm Association in 20 European countries after the COVID-19 pandemic, remote monitoring of pacemakers increased from 24% to 40%, and within this, only 60% of the work was effectively registered, creating a scenario of reduced accessibility and insufficient financial sustainability [29]. On the other hand, these patients are often elderly and highly dependent, sometimes requiring reimbursement for non-emergency transport, one of the main barriers to healthcare. In this context, remote monitoring is a follow-up alternative that helps to reduce travel and transport, leading to a reduction in face-to-face visits [26,30]. In addition, remote monitoring costs, especially equipment costs, are one of the limiting factors, as the cost–benefit ratio for this group of users is still controversial [29,30,31].

In Portugal, patients can request reimbursed non-emergency public transport based on economic and clinical criteria. The factors that influence the cost of transport are price per km, the need for a caregiver, the type of vehicle used, the existence of additional care required (e.g., oxygen), waiting times, and the percentage of reimbursement depending on whether there is shared transport. Healthcare institutions pay full reimbursement to the most distant patient, called the “1st user.” If there is no shared transport, this user is always the “1st user” (with maximum reimbursement). If there is shared transport, the patient or patients with shorter distances are called “2nd user,” and therefore, hospitals and other institutions contribute with a lower co-payment (up to 30%, depending on the distance) [32,33,34,35,36,37]. In 2023 alone, more than 2.3 million subsidized transports were carried out across the country, representing an all-time high of almost EUR 81 million [38].

Few studies have focused on reimbursed transport in this group of patients, the impact on healthcare services, and the possibility of offsetting this cost with remote technology while increasing safety and improving healthcare. Therefore, there is a need to evaluate follow-up protocols, criteria, and new opportunities for financial sustainability regarding remote follow-up equipment and service [39].

The aim of this study was to assess the financial impact of the costs associated with reimbursed transport versus the investment in remote monitoring equipment for pacemaker patients in a Portuguese district hospital and to evaluate the acceptance and satisfaction of this follow-up alternative. A secondary objective was to assess the impact of age and distance on the acceptance of this technology (Figure 1).

## 2. Materials and Methods

This was an exploratory, quantitative, cross-sectional study with convenience sampling based on a prior universe analysis of patients with a pacemaker, in a portuguese district hospital. A minimum sample size of 38 patients was calculated using Cochran’s formula, considering a 95% confidence interval (C.I.) for a finite population of 2100 patients. Based on previous data, the proportion of patients with pacemaker-reimbursed transportation was 2.6% [40]. Regarding sample collection, the inclusion criteria were patients undergoing pacemaker follow-up who were beneficiaries of reimbursed non-urgent transport and who agreed to participate, with informed consent signed. Patients under 18 years old were excluded.

Questionnaires were administered to patients or, whenever this proved to be impossible, to their legal representatives. Informed consent was obtained from all subjects involved in the study. To evaluate the results, a hypothetical remote follow-up protocol was simulated based on existing recommendations and considering similar costs between each face-to-face visit. This evaluation focused on the costs of remote communication as opposed to the costs associated with non-urgent transportation. No additional factors were tested. The simulated protocol was as follows:Classic follow-up: In-office appointments every 6 months.Remote follow-up: One in-office visit (every 2 years) and three remote consultations every 6 months (Figure 2).

Regarding pacemaker battery longevity, the calculation was based on the implantation date and the remaining battery life measured during consultation, which also allowed the chronology of the protocols tested to be adjusted on a case-by-case basis. The cost of the equipment needed for remote monitoring and the cost of reimbursement for non-emergency transport were considered, based on the number of kilometers traveled, the presence of medical professionals or relatives, and waiting times measured in a questionnaire. This assessment was based on Portuguese legislation [32,33,34,35,36,37,41]. Since it was impossible to accurately assess the context of shared transport for all users, different scenarios were considered. In the first scenario, all participants were considered as “1st users” (with full reimbursement) for each type of transport (ambulance or other vehicle), and the sample was then divided according to the type of transport. Sub-analyses were carried out for distances ≥30 km from the hospital (≥60 km round trip). In the case of shared transport (‘2nd users’), partial reimbursement was considered as 30% paid between 15 and 30 km, 20% paid between 30 and 100 km, and 15% paid for distances over 100 km. Due to sample dispersion, no other scenarios were predicted. After 2026, the price per kilometer will remain unchanged according to the last update provided in legislation no. 7606/2023. Finally, to increase accuracy, an additional 10% travel distance was assumed to account for routes. Transport costs measured in the questionnaire, as well as the costs associated with the remote monitoring equipment, were calculated for each condition previously described. For each patient and for each condition described, a three-step approach was conducted:

Step 1: Assessment of transportation-related costs and protocol conditions for each patient (Table 1).

Step 2: Protocols simulation for each patient (Table 2).

Step 3: Cost comparison (Figure 3).

After calculating the differences between the costs of the two protocols for all patients, statistical inference tests were performed. EZR, version 1.61, was used. Administrative and personnel costs variables were considered constant and were therefore not considered. No additional price adjustments were made, as this is imposed by public legislation.

## 3. Results

Questionnaires were collected from 41 patients between 56 and 100 years old (83 on average). The majority were female (68.3%) compared to 31.7% males. In terms of education, the majority had an elementary school education (63.5%), followed by secondary and higher education (2.4% each). Patients in this sample who did not attend any type of school represented 37%.

The distances traveled ranged from 15 km to 280 km, representing the longest travel distance. The overall average distance traveled was approximately 81 km.

In this study, the average battery life of the devices was 12 years (minimum 8 years and maximum 15 years).

From a clinical standpoint, only 8 of the 41 participants were independent, while 34.2% were partially dependent and 46.3% were totally dependent.

Regarding the type of vehicle requested, the majority used an ambulance (63%) compared to other non-emergency patient transport vehicles (37%).

In this sample, 66% of the patients did not share transport (1st user), while the remaining 34% shared transport with one or more patients. For 21 (81%) of the 26 ambulances requested, there was no shared transport. Specifically, for users who needed a stretcher to be moved, there was no shared transport in 85% of cases, and they were therefore necessarily considered as “1st user.” Regarding other vehicles, only 40% were considered for full reimbursement. Table 3 shows the distribution of the type of vehicle with shared transport.

### 3.1. Summary of Expenses for the Simulated Protocols

Implementation of a remote follow-up protocol in patients with a pacemaker showed statistically significant savings in all scenarios studied, considering full reimbursement. Savings were higher when a minimum distance of 30 km (60 km round trip) was considered. Globally, savings of EUR 373 were calculated (*p* = 0.01151) for full reimbursement, increasing to EUR 625 for ≥60 km (round trip), with strong statistical evidence (*p* = 0.000001624). Considering only ambulances with full reimbursement, savings of EUR 422 were projected (*p* = 0.0312), increasing to EUR 764 with ≥60 km of travel distance, with strong statistical evidence (*p* = 0.0002747). Regarding other vehicles, savings of EUR 270 were calculated (*p* = 0.042), achieving EUR 439 for ≥60 km round trip, with strong statistical evidence (*p* = 0.000977). On the other hand, considering only shared transport patients, there was a simulated loss of EUR 575 (*p* = 3.82 × 10^−11^). Considering all patients with reimbursement (total or partial) and any type of vehicle, savings of EUR 64 were calculated, but without statistical significance (*p* = 0.21). No other scenarios were considered.

Table 4 summarizes all calculations resulting from the simulated trial protocols.

### 3.2. Reduction in Face-to-Face Consultations

The protocol under study resulted in a reduction in face-to-face visits from approximately 26 to 7 consultations on average, corresponding to an approximately 74% reduction in hospital visits (*p* = 0.0000000229).

### 3.3. Acceptance and Satisfaction with Remote Monitoring

Of the 41 patients, 88% would accept remote monitoring, having answered “totally agree” (51%) or “agree” (37%), while 12% maintained their preference for regular hospital visits. Regarding satisfaction, 88% of the participants responded positively to remote follow-up, interspersed with face-to-face hospital appointments.

Concerning age (*p* = 0.103) and distance (*p* = 0.737) factors, there was no relationship with remote follow-up acceptance.

## 4. Discussion

There is a gap in the research relating to anti-bradycardia pacemakers and the financial viability of remote monitoring. Furthermore, available studies cannot be extrapolated due to multi-factorial conditions, such as geographic characteristics, political policies, and country-related financial conditions.

The clinical and technical benefits associated with remote monitoring are well supported in the literature, and patients with more complex devices benefit the most. However, there is still controversy from a cost–benefit perspective regarding the widespread adoption of this technology for patients with pacemakers [26]. Cost savings for patients and caregivers, combined with savings related to non-urgent transportation supported by public reimbursement, may represent a window of opportunity for remote monitoring. This study focused on equipment and transportation costs, one of the main barriers to healthcare access.

The simulated protocols measured follow-up savings in patients with pacemakers who used reimbursed non-emergency transportation during the expected battery life of the device (12 years on average in this sample). According to Portuguese legislation, one of the most impactful factors on the payment for travel distance is the “1st user” condition (the most distant patient—full reimbursement) versus “2nd user” (other patients in the case of shared transport) [36,41]. Considering all patients as “1st user,” the reduction in reimbursed transportation costs exceeded the initial costs needed for remote monitoring equipment, resulting in 21.2% savings. In contrast, considering patients with shared transport, the expected cost reduction was not enough to cover the initial investment, resulting in an estimated loss of EUR 575, even for maximum distances, and no further analysis was carried out regarding this aspect. When the sample was globally evaluated considering any reimbursement and mixed vehicle type, the savings were not significant (*p* = 0.21), emphasizing the need to establish criteria that ensure the financial viability of remote monitoring. This can be explained by the different percentages of reimbursement previously mentioned.

Published legislation also distinguishes between the types of vehicles required. Specifically, there were projected savings of 23% for ambulances and 16.7% for other vehicles, considering full reimbursement. The collected data allowed us to identify the travel distance from which the savings from reimbursed transportation exceeded the initial investment in remote monitoring equipment (considering full reimbursement). For distances to the hospital ≥30 km (≥60 km round trip), the simulated savings reached 33.3% (EUR 764) for ambulance users and 23.9% (EUR 439) for other vehicles. The purpose of this secondary evaluation was to increase the sensitivity of the conclusions by suggesting attribution criteria of remote monitoring equipment that would guarantee financial viability with a very strong statistical correlation.

The majority of patients (85%) who requested an ambulance and required a stretcher did not share transport, being automatically considered the “1st patient,” so there was a higher chance for savings. On the other hand, when using other vehicles, shared transport was more frequent (60%). This fact makes it impossible to include such a condition as financially viable, because of the inability to determine travel distances for all patients. Therefore, it is not possible to determine the reimbursement percentage.

It is difficult to make a direct comparison with the results of other national or international studies, either due to the type of equipment analyzed, the different geographical distances, the financing model for public services, or, in this specific case, the costs of transportation, equipment, and technology. Therefore, it is not possible to establish a cross-section of the results obtained. Most studies outside the Portuguese context focus on costs from the perspective of users and their families. The ECOST study, which analyzed individual transport costs, estimated an overall saving of up to 24% (with weak statistical correlation), a value in line with the 21.2% projected for the whole sample [39].

At the national level, there are references to potential savings from reimbursed transportation, but no specific figures have been found that combine the impact of these costs with the reduction in hospital visits in the context of telemonitoring [42].

Other factors may affect the cost analysis. Most remote monitoring devices on the market are reusable. Therefore, even in the event of device replacement, withdrawal, or eventual death, the initial investment would not be lost and could even provide greater savings in the long run. The possibility of equipment reusability, as well as warranty conditions, should be taken into account when selecting devices. In addition, as battery performance improves and average life expectancy increases, savings may be even greater in the future.

The reduction in hospital visits was the main consequence of the remote monitoring, which allowed a reduction in reimbursed transportation expenses. The application of the remote protocol would allow an average reduction in face-to-face visits from around 26 to 7, which corresponds to a reduction of almost 74% in hospital visits. This number is close to the best results found in the literature. Protocols that rely heavily on remote monitoring have achieved reductions in face-to-face visits of up to 81% [43,44,45].

In a national context, this result is higher than the results of the PORTlink study. This study found a 67.8% reduction in face-to-face visits [46]. This difference can be explained by the types of devices evaluated. In the case of simpler, automatic pacemakers, the need for in-person corrections and reprogramming may be lower compared to more complex devices in patients who tend to be younger but often have severe pathology [16].

This study found a high acceptance and satisfaction rate (88%) with remote monitoring. There is a general perception that this technology contributes to regular and safer patient monitoring while significantly reducing in-office visits. It is important to note that this assessment was based on an explanation of how remote consultations are processed, but not on previous experience. This may explain a slightly lower percentage than other studies, which found a 97% satisfaction rate with this type of follow-up [5]. In Portugal, in the largest recent randomized trial, 95% of participants found the remote equipment easy or very easy to use [46].

Another relevant aspect is the perception of patients and carers about health services. In this research, 90.2% of respondents believed that these non-face-to-face consultations are easier for organizations and professionals, which is in line with other research that reported 90% on this topic [5]. While it is true that there is potential for this to happen, it is also true that organizations are still heavily based on face-to-face services, and there is still a need for profound reorganizations so that remote technology does not become an additional burden but rather a real alternative [16].

In terms of acceptance of this technology, there was no correlation with age or travel distance. The need for healthcare plays a key role that may not be present in other contexts. In other words, regardless of the kilometers traveled, the benefit of this type of consultation was recognized, pointing to the fragility of the users in the sample studied. Likewise, while in other contexts there may be a greater aptitude for technology among younger ages, this was not proven here, as described by other authors [5,47,48].

The criteria studied can be replicated by other organizations if the conditions are met. The variables addressed, namely transportation and equipment costs, are the result of regulations and public legislation that can be evaluated and compared. Based on the discussion presented and for the conditions analyzed, it is possible to propose criteria for the allocation of remote monitoring equipment for pacemaker follow-up, with financial viability, under the following assumptions:Type of transportation: Ambulance, patient on stretcher (high probability of being “1st patient”—full reimbursement);Distance to health institution: ≥30 km (corresponding to ≥60 km round trip);Expected follow-up: 12 years (including pacemaker replacements and equipment reuse).

Under these conditions, all participants obtained significant savings with strong statistical evidence in this protocol simulation. Of the sample collected, 85% of the participants who requested an ambulance and needed a stretcher for transportation did not share transport and therefore obtained the maximum reimbursement. Even if there are some cases in which the “1st patient” criterion is not met, overall, there may be a positive financial balance. Considering other vehicles, there is a high probability of shared transportation, so the financial viability of equipment costs is not guaranteed.

### Limitations to the Study

This study focused on the costs associated with the non-emergency transportation of patients compared to the costs associated with remote monitoring equipment and did not consider other direct or indirect savings, as this was not the focus of this investigation.

The results of this study were achieved by extrapolating the information obtained according to the hypothetical follow-up protocols established. Also, the reduced sample size must be considered when evaluating the results. The protocols created were a combination of existing clinical evidence and the usual practice of the district hospital under study, and there may be differences for other institutions. As such, this research is seen as an initial assessment of the financial viability of remote consultations for patients with pacemakers in this hospital and may not be representative in all scenarios. In addition, some of the answers to the questionnaire were derived from the explanation given about remote monitoring and not from experience already acquired. Some aspects related to travel distances and transport are difficult to measure, such as the exact departure and arrival sites, as well as possible route deviations or even simultaneous transport services. Even so, this factor will ultimately contribute to a possible undervaluation of the results obtained, without detriment to the conclusions presented.

From another perspective, industry organization and the variability of supply and demand for medical equipment, associated with the specific nature of healthcare services, make it highly difficult to predict future costs. Therefore, it was decided to measure the current average cost of the remote monitoring equipment. On the other hand, the costs associated with transportation were based on existing, past, and future legislation (until 2026). Since the values are stipulated by law, it was decided not to reflect inflation adjustments after that date.

## 5. Conclusions

From a resource management perspective, the cost savings associated with fewer in-person visits and, therefore, with lower reimbursed transportation, could cover the investment in equipment associated with remote monitoring. From a healthcare services perspective, it is an opportunity to improve patient care capacity and quality of life without increasing costs. To achieve these objectives, it is important to define what criteria lead to financial sustainability. Remote monitoring in this study population is considered financially viable for patients with pacemakers who use reimbursed transportation for an expected follow-up period of approximately 12 years, as the “1st patient” (with full reimbursement), while savings are more significant for patients who use an ambulance. Additionally, regardless of the type of vehicle, more significant savings were found for distances ≥30 km (≥60 km round trip).

In summary, remote monitoring of patients with simple anti-bradycardia pacemakers, requiring reimbursed transportation, may be financially viable and is also associated with a high percentage of acceptance and satisfaction.

### Future Investigations

In alignment with existing recommendations and the results obtained, future studies should consider a prospective and multicenter approach to investigate reimbursement and accounting mechanisms for these services within the framework of current legislation. Additionally, it would be valuable to assess the impact of user satisfaction and engagement on remote monitoring management and human resources efficiency. Simultaneously, exploring new sources of economic and financial inefficiencies in healthcare policies and systems could reveal financial opportunities to enhance accessibility to this technology.

## Figures and Tables

**Figure 1 healthcare-13-03257-f001:**
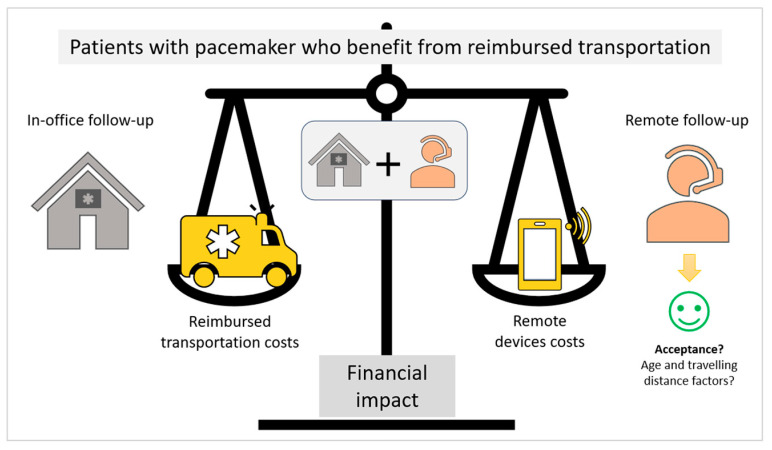
Overall study objectives.

**Figure 2 healthcare-13-03257-f002:**
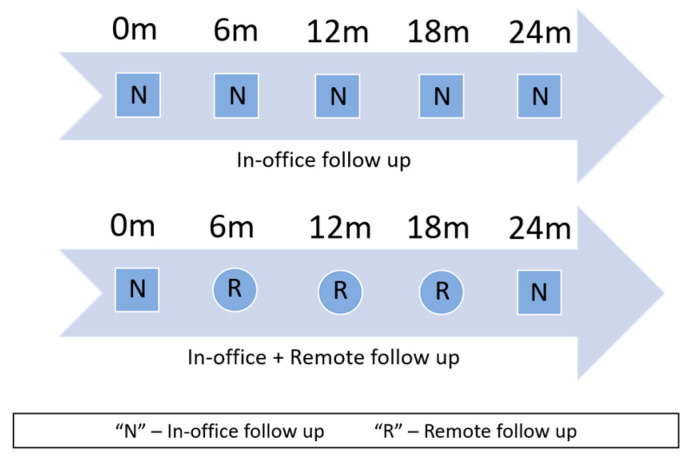
Simulated follow-up protocol for 24 months (m—months).

**Figure 3 healthcare-13-03257-f003:**
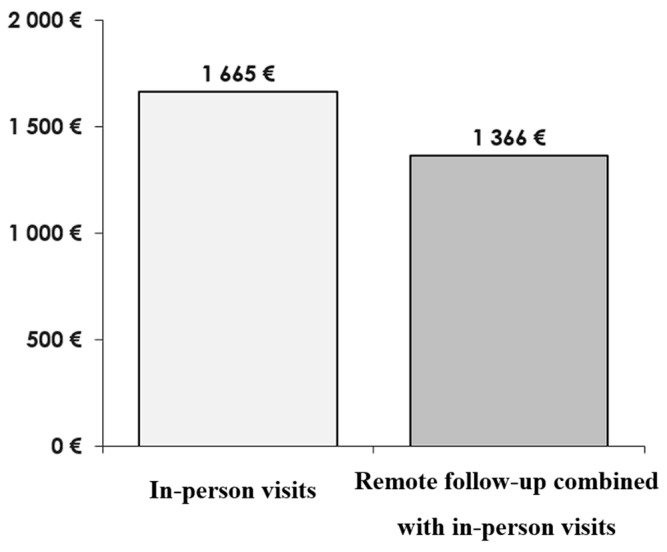
Example of cost comparison between the simulated protocols.

**Table 1 healthcare-13-03257-t001:** Example of transportation-related costs and protocol conditions for one patient.

Year	Cost per km (€)	Transport Cost (€)	Waiting Time Cost (€)	Caregiver Cost (€)
**2012–2021**	0.51	30.60	5.00	3.06
**2022–2023**	0.58	34.80	10.00	3.48
**2024**	0.63	37.80	11.50	5.39
**2025**	0.66	39.60	12.13	5.60
**>2026**	0.69	41.40	12.77	5.82
Personal costs are considered constant. Example: 13-year total device battery life; ambulance; 60 km travel distance. Average cost of the device for remote consultation: €943.41 (including VAT)				

**Table 2 healthcare-13-03257-t002:** Tested protocols example considering full reimbursement.

**In-person visits (N) protocol simulation**																
**Costs**				**1 Year**		**2 Years**		**3 Years**		**4 Years**		**5 Years**		**6 Years**		**7 Years**
**Item**		**N1**	**N2**	**N3**	**N4**	**N5**	**N6**	**N7**	**N8**	**N9**	**N10**	**N11**	**N12**	**N13**	**N14**	**N15**
Remote device cost		0 €	0 €	0 €	0 €	0 €	0 €	0 €	0 €	0 €	0 €	0 €	0 €	0 €	0 €	0 €
Transport cost		34.80 €	34.80 €	34.80 €	34.80 €	37.80 €	37.80 €	39.60 €	39.60 €	41.40 €	41.40 €	41.40 €	41.40 €	41.40 €	41.40 €	41.40 €
Waiting time cost		10.00 €	10.00 €	10.00 €	10.00 €	11.50 €	11.50 €	12.13 €	12.13 €	12.77 €	12.77 €	12.77 €	12.77 €	12.77 €	12.77 €	12.77 €
Caregiver cost		3.48 €	3.48 €	3.48 €	3.48 €	5.39 €	5.39 €	5.60 €	5.60 €	5.82 €	5.82 €	5.82 €	5.82 €	5.82 €	5.82 €	5.82 €
Additional 10%		3.48 €	3.48 €	3.48 €	3.48 €	3.78 €	3.78 €	3.96 €	3.96 €	4.14 €	4.14 €	4.14 €	4.14 €	4.14 €	4.14 €	4.14 €
Total		51.76 €	51.76 €	51.76 €	51.76 €	58.47 €	58.47 €	61.29 €	61.29 €	64.13 €	64.13 €	64.13 €	64.13 €	64.13 €	64.13 €	64.13 €
	**8 Years**		**9 Years**		**10 Years**		**11 Years**		**12 Years**		**13 Years**		**14 Years**		**15 Years**	
**N16**	**N17**	**N18**	**N19**	**N20**	**N21**	**N22**	**N23**	**N24**	**N25**	**N26**	**N27**	**N28**	**N29**	**N30**	**N31**	**Total**
0 €	0 €	0 €	0 €	0 €	0 €	0 €	0 €	0 €	0 €	0 €	0 €					0 €
41.40 €	41.40 €	41.40 €	41.40 €	41.40 €	41.40 €	41.40 €	41.40 €	41.40 €	41.40 €	41.40 €	41.40 €					1080.60 €
12.77 €	12.77 €	12.77 €	12.77 €	12.77 €	12.77 €	12.77 €	12.77 €	12.77 €	12.77 €	12.77 €	12.77 €					329.89 €
5.82 €	5.82 €	5.82 €	5.82 €	5.82 €	5.82 €	5.82 €	5.82 €	5.82 €	5.82 €	5.82 €	5.82 €					146.48 €
4.14 €	4.14 €	4.14 €	4.14 €	4.14 €	4.14 €	4.14 €	4.14 €	4.14 €	4.14 €	4.14 €	4.14 €					108.06 €
64.13 €	64.13 €	64.13 €	64.13 €	64.13 €	64.13 €	64.13 €	64.13 €	64.13 €	64.13 €	64.13 €	64.13 €					1665 €
**Remote follow-up (R) combined with in-person visits (N) protocol simulation**																
**Costs**				**1 Year**		**2 Years**		**3 Years**		**4 Years**		**5 Years**		**6 Years**		**7 Years**
**Item**		**N1**	**R2**	**R3**	**R4**	**N5**	**R6**	**R7**	**R8**	**N9**	**R10**	**R11**	**R12**	**N13**	**R14**	**R15**
Remote device cost		935.00 €				0 €				0 €				0 €		
Transport cost		34.80 €				37.80 €				41.40 €				41.40 €		
Waiting time cost		10.00 €				11.50 €				12.77 €				12.77 €		
Caregiver cost		3.48 €				5.39 €				5.82 €				5.82 €		
Additional 10%		3.48 €				3.78 €				4.14 €				4.14 €		
Total		986.76 €				58.47 €				64.13 €				64.13 €		
	**8 Years**		**9 Years**		**10 Years**		**11 Years**		**12 Years**		**13 Years**		**14 Years**		**15 Years**	
**R16**	**N17**	**R18**	**R19**	**R20**	**N21**	**R22**	**R23**	**R24**	**N25**	**R26**	**R27**	**R28**	**N29**	**R30**	**R31**	**Total**
	0 €				0 €				0 €							935 €
	41.40 €				41.40 €				41.40 €							279.60 €
	12.77 €				12.77 €				12.77 €							85.35 €
	5.82 €				5.82 €				5.82 €							37.97 €
	4.14 €				4.14 €				4.14 €							27.96 €
	64.13 €				64.13 €				64.13 €							1366 €

**Table 3 healthcare-13-03257-t003:** Relationship between “1st and 2nd user” (full and partial reimbursed transport) and type of vehicle.

	Ambulance	Other Vehicles	Total
**Full reimbursement** n (%)	21 (51%)	6 (15%)	27 (66%)
**Partial reimbursement** n (%)	5 (12%)	9 (22%)	14 (34%)
**Total** n (%)	26 (63%)	15 (37%)	41 (100%)
Fully reimbursed patients using ambulance = 81%			
Partially reimbursed patients using ambulance = 19%			
Fully reimbursed patients using other vehicles = 40%			
Partially reimbursed patients using other vehicles = 19%			
Full reimbursed patients using ambulance + stretcher = 85%			

**Table 4 healthcare-13-03257-t004:** Expected average difference in costs between protocols over the expected longevity of the devices, considering a 95% confidence interval (C.I.).

	n	In-Office Follow-Up Average (EUR)	In-Office + Remote Follow-Up Average (EUR)	Difference (EUR)	*p*
All patients; full reimbursement	41	EUR 1757	EUR 1384	EUR 373 (21.2%)	0.01151
All patients; ≥60 km; full reimbursement	29	EUR 2101	EUR 1476	EUR 625 (29.8%)	0.000001624
Ambulances only; full reimbursement	26	EUR 1828	EUR 1406	EUR 422 (23.1%)	0.01151
Ambulances only; ≥60 km; full reimbursement	17	EUR 2292	EUR 1528	EUR 764 (33.3%)	0.0002747
Other vehicles: full reimbursement	16	EUR 1613	EUR 1343	EUR 270 (16.7%)	0.042
Other vehicles; ≥60 km; full reimbursement	12	EUR 1838	EUR 1399	EUR 439 (23.9%	0.000977
Patients with partial reimbursement	18	EUR 633	EUR 1101	−EUR 735 (−48.2%)	0.00000000133
All patients; any reimbursement/vehicles	41	EUR 1345	EUR 1281	EUR 64 (4.8%)	0.21

95% C.I.

## Data Availability

Due to data protection laws, the data presented in this study are only available upon request in justified cases and after agreement of the investigation department of Santarém District Hospital.

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
