# Peer review of "Comparison Between In-Office Versus Remote Follow-Up Costs in Patients with Pacemakers and Reimbursed Transportation in a Portuguese District Hospital"

_healthcare, 2025, doi:10.3390/healthcare13243257_

Round 1

Reviewer 1 Report

Comments and Suggestions for Authors

Abstract

In the Abstract, clearly state that cost analysis is simulated rather than based on observed remote follow-up.

Introduction

It is well structured, and the research question is fully addressed. The literature review has been conducted thoroughly.

Materials and Methods

Sample Size Justification Requires Clarification. The author used a proportion of 2.6%; it is not adequately clear why this proportion was selected, also how the sample was selected.

Costs related to equipment, battery longevity, waiting time, and transport are considered the same in the follow-up years. Small changes may significantly alter cost-effectiveness.

What were the inclusion and exclusion criteria for the study?

Some cost components were not included in the study, such as IT infrastructure and licensing, training requirements, and staff workload. All Participants experienced remote monitoring for evaluation of their acceptance.

Results

At the end of the first paragraph,” Patients in this sample, who did not attend any type of school, represented 37%”. That means they were illiterate?

The main cost-saving results are driven by assuming that all individuals are “1st users.” However, only 66% of the sample actually met this criterion. Provide a weighted real-world cost analysis using the actual reimbursement distribution in the sample

Include confidence intervals alongside p-values.

Discusion

It has a good structure, but the limitations of the study, such as low sample size and its representation of the real society, should be further pointed out.

- Minor English language revisions are needed for clarity and readability.

Author Response

First of all, Thank you for your considerations an suggestions. Please consider my comments in "red".

1- In the Abstract, clearly state that cost analysis is simulated rather than based on observed remote follow-up.

  Sentence change:  The projected costs were calculated for two simulated protocols..

2 - Sample Size Justification Requires Clarification. The author used a proportion of 2.6%; it is not adequately clear why this proportion was selected, also how the sample was selected.

There are defined criteria for awarding reimbursed transport. Based on information from the previous year, the proportion of users with publicly paid transport in relation to the total population of users with a pacemaker was 2.6%. I will clarify in the text.

3 - Costs related to equipment, battery longevity, waiting time, and transport are considered the same in the follow-up years. Small changes may significantly alter cost-effectiveness.

In this case, a conservative approach was chosen. Being a protocol simulation and not a prospective study, it would be impossible to accurately estimate all costs throughout follow-up. Yes, waiting time was considered constant. However, the longevity of the devices was carefully checked for each user and transport was also calculated individually using the available legislation, assuming that there would be no changes to the place of residence. The cost of the equipment is a one-time purchase and was based on current values.

 4- What were the inclusion and exclusion criteria for the study?

changes:
Regarding sample collection, as inclusion criteria, were considered patients undergoing a pacemaker follow-up, beneficiaries of reimbursed non-urgent transport, who agree to participate and with informed consent signed. Patients under 18 years old were excluded.

5 - At the end of the first paragraph,” Patients in this sample, who did not attend any type of school, represented 37%”. That means they were illiterate?

Yes 

6 - The main cost-saving results are driven by assuming that all individuals are “1st users.” However, only 66% of the sample actually met this criterion. Provide a weighted real-world cost analysis using the actual reimbursement distribution in the sample

We tottaly agree with you. This is the reason why we emphasize the more specific result for ambulances because in this case the maximum contribution is much more frequent, reaching up to 85%. But we understand your point and we provided aditional clarifications in this regard in the results and discussion. We ran data again.

.. When the sample was globally evaluated considering any reimbursement and vehicle type, the savings were not significant (p=0.21), emphasizing the need to establish criteria that ensure the financial viability of remote monitoring in this context. This can be explained by the different percentages of reimbursement previously mentioned.

7- It has a good structure, but the limitations of the study, such as low sample size and its representation of the real society, should be further pointed out.

Sentences changes:

...Also, the reduced sample size must be taken into account when evaluating the results. The protocols created were a combination of existing clinical evidence and the usual practice of the district hospital under study, and there may be differences for other institutions. As such, this research is seen as an initial assessment of the financial viability of remote consultations for patients with pacemakers in this hospital and may not be representative in all scenarios

Reviewer 2 Report

Comments and Suggestions for Authors

Congratulations for the research work. However, some observations have to be made regarding the subject, structure, research methodology, data analysis and results as it follows :

The title of the research work Comparison between in office versus remote follow up costs in patient with peace maker and reimbursed transportation in a Portuguese District is adequate to the content

The abstract presents in short the research. However, the research methods are not full disclosed, besides the mention of surveys number which is a little bit low (41) for a research study.  Therefore, is advisable to mention them. The computer software and the method used in research as well.

The key words are ok

The introduction argue the purpose of research. The  main argument and the novelty pf the research idea,  is that the distinguished authors propose to analyse "the financial impact of the costs associated with the reimbursed transport versus the investment in remote monitoring equipment for pace makers in Portuguese Hospitals.

In the introduction there is no references about the research questions and research hypothesis, there are usually considered in research papers. Will be a good idea to add them.

Literature review There is not presented distinctively as an research article's part. It is advisable to add it.

Materials and methods-In this part  the distinguished authors  presents the Research Methodology from collecting data from 2100 patients. Here I am a little bit confused because the distinguished authors stated in the abstract that they used data from 41 surveys. Some explanations on this subject will be useful. A very accurate cost analysis with graphic models, regarding heart patient transportation was performed by the authors in the present research.

The static analysis performed in the study  and data interpretation is also good. The problem is that the analyse is taking into consideration a  low number of respondents (41) .

Discussions and data interpretation are good

Conclusions are relevant for the study

The List of References is adequate and sustain the research purpose.

Author Response

First of all, thank you for your time and considerations. Answers in red..

1 - The abstract presents in short the research. However, the research methods are not full disclosed, besides the mention of surveys number which is a little bit low (41) for a research study.  Therefore, is advisable to mention them. The computer software and the method used in research as well.

A total of 41 surveys were conducted with patients who received a pacemaker and used public reimbursed transportation. The projected costs were calculated for two simulated protocols: the first involved in-person visits every six months, while the second involved in-person visits every two years with remote consultations every six months, over the expected lifespan of the devices. EZR version 1.61 was used.

2 - Literature review There is not presented distinctively as an research article's part. It is advisable to add it.

I followed the provided template which has this structure. I also checked several articles published in this magazine and they all follow this structure. We also didn't want to have a very long article. Is there any chance you can allow it to stay like this?

3 - Materials and methods-In this part  the distinguished authors  presents the Research Methodology from collecting data from 2100 patients. Here I am a little bit confused because the distinguished authors stated in the abstract that they used data from 41 surveys. Some explanations on this subject will be useful.

2100 refers to the total number of users with anti bradycardia pacemaker. Only around 2.6% had access to reimbursed transport according to previous information. The minimum sample would be 38 patients. We got 41. Clarifications were made regarding this aspect.

Other changes were made to clarify results and conclusions. Thank you

Reviewer 3 Report

Comments and Suggestions for Authors

This is just a repetition of the following paper:

https://www.sciencedirect.com/science/article/pii/S2174204913000500?via%3Dihub

Author Response

Thanks for the comment.

The article in question is just an old literature review bringing together information from several studies with all types of devices. There is no specific relationship with the transport factor, which is one of the main focuses of our study. Our study focuses only on pacemakers and publicly reimbursed transportation. There is no overlap with the mentioned article.

If you need any more information, please let us know.

Best regards.

Round 2

Reviewer 3 Report

Comments and Suggestions for Authors

OK